# Reproductive, maternal, newborn, child and adolescent health services in humanitarian and fragile settings: A mixed methods study of midwives' and women's experiences

T. Dey[1]*, M. G. Shah[1], A. Baba[2], N. Mugo[3], T. Thommesen[4], V. Vivilaki[5], M. Boniol[6], N. Alam[7], M. Dibley[7], D. Okoro[8], P. Tenhoope-bender[8], T. Triantafyllou[9], E. V. Langlois[1]

1 Partnership for Maternal, Newborn and Child Health, Geneva, Switzerland, 2 Institut Panafricain de Santé Communautaire, Aru, Democratic Republic of Congo, 3 NSW Health, Priority population Unit, Integrated and community health, Cumberland Hospital, New South Wales, Sydney, Australia, 4 Stavanger University Hospital, Stavanger, Norway, 5 International Confederation of Midwives (ICM), The Hague, The Netherlands, 6 World Health Organization (WHO), Geneva, Switzerland, 7 University of Sydney, Sydney, Australia, 8 United Nations Population Fund (UNFPA), New York, United States of America, 9 University of West Attica, Athens, Greece

* teesta.dey@liverpool.ac.uk

**Data Availability Statement:** This was a secondary analysis of the following existing sources: 'Being a

## Abstract

Insufficient progress has been made to reduce morbidity and mortality for women, children and adolescents particularly in Humanitarian and Fragile settings (HFS). Midwives play a critical and unique role in ensuring communities receive quality and safe essential sexual, reproductive, maternal, newborn, child, and adolescent health services. A lack of knowledge exists on the availability and experiences of midwifery services in HFS. This manuscript provides an overview of the midwifery density in HFS and a synthesis of the experiences of women receiving midwifery care, and barriers and facilitators for midwives providing essential SRMNCAH services in HFS. Guided by an expert committee, a concurrent mixed methods approach was applied, using secondary analysis of primary quantitative and qualitative data sources. Quantitative analysis of the global distribution of midwives compared to fragility was undertaken. Qualitative analysis of experiences of receipt and provision of midwifery care was undertaken across four settings providing humanitarian care. There is a critically low density of midwives in humanitarian and fragile settings. Sub-Saharan Africa accounts for the highest levels of fragility yet lowest density of midwives able to provide SRMNCAH services. Lack of finances both constrains midwives from effectively providing services and prevent communities from utilising services. Sub-optimal working conditions through rising workloads, insufficient and/or inconsistent resources were frequently reported to impede midwives from providing care in HFS. Uniquely for HFS, threats to the safety and security of midwives to conduct their work was widely reported. Key facilitators identified included, complex adaptive health system designs to respond effectively to the rapidly changing HFS environment, realisation of supporting "power, agency and status" as instrumental for midwives to provide quality care and promotion of community-centric approaches may enable continuity of care and uptake of essential SRMNCAH services. Midwives are critical to

midwife is being prepared to help women in very difficult conditions': midwives' experiences of working in the rural and fragile settings of Ituri Province, Democratic Republic of Congo Baba, Theobald, et al. 2020 "The midwife helped me, otherwise I could have died": women's experience of professional midwifery services in rural Afghanistan - a qualitative study in the provinces Kunar and Laghman Thommesen et al. 2020 Midwives' experiences of cultural competency training and providing perinatal care for migrant women a mixed methods study: Operational Refugee and Migrant Maternal Approach (ORAMMA) project Fair et al. 2021 Barriers Faced by the Health Workers to Deliver Maternal Care Services and Their Perceptions of the Factors Preventing Their Clients from Receiving the Services: A Qualitative Study in South Sudan Mugo et al. 2018

**Funding:** The authors received no specific funding for this work.

**Competing interests:** The authors have declared that no competing interests exist.

protect the health and well-being of communities. They require urgent protection and prioritisation in HFS areas where the need is greatest.

## Background

Considerable progress has been made over the past decades in reducing maternal and newborn mortality [1, 2]. However, progress has not been achieved evenly. In particular, emerging data show that women, children, and adolescents living in humanitarian and fragile settings (HFS) bear a disproportionate risk of mortality and morbidity. For instance, women of reproductive age living near the highest intensity conflicts have three times higher mortality than women living in peaceful settings [3]. Similarly, 43% of global under-five deaths in 2020 occurred in conflict and fragile affected countries and nine of 10 countries with the highest neonatal mortality rates are in conflict [4, 5].

In HFS, weak infrastructures, high levels of insecurity, population movement, disruptions in supply chains, and exacerbations of pre-existing shortages of human and financial resources, often occur [6]. It is therefore especially difficult to deliver even the most essential of services to affected populations, particularly women, children, and adolescents. The challenges are particularly dire outside of camp settings where services are less accessible through fixed health facilities.

Additionally, there is a stark intersectionality between various vulnerabilities in HFS including but not limited to poverty, displacement, and gender inequality. In these settings, women and girls are far more likely to experience gender-based violence and are extremely vulnerable to physical, sexual, and psychological harm [7]. For instance, violence against women and girls accounted for 97% of conflict-related sexual violence cases reported in 2021 [8]. A review of 19 studies across 14 countries estimated that about 21% of displaced women experienced sexual violence [5]. Up to a third of girls living in a humanitarian setting report that their first sexual encounter was forced [9]. These inequities have been further compounded by the Covid-19 pandemic and climate change, linked to drought and food insecurity. At the height of the pandemic, up to a third of essential sexual, reproductive, maternal, newborn, child, and adolescent health (SRMNCAH) services were disrupted, including in humanitarian settings [10].

In such contexts, midwives play a critical role in ensuring that women and their babies are receiving adequate, quality, and continuum of care during and after pregnancy and childbirth. Due to their knowledge and skills as frontline health-care providers and their geographic and social proximity to the communities they serve, midwives can reach the most vulnerable and hard-to-reach women and their newborns [11].

Considering the close link they have to the communities, midwives are also uniquely placed to provide a full range of SRMNCAH people-centred care services beyond immediate maternal and newborn care [12]. This is especially important in HFS where health systems are broken or fragile and access is inconsistent. It is estimated that midwives can provide 90% of essential SRMNCAH services when supported in an enabling environment necessary to scale midwife-led care. As such are critical for optimum, cost-effective service delivery [11]. In addition to preventing maternal and newborn deaths, quality midwifery care can also improve over 50 other health-related outcomes, including—sexual and reproductive health, immunisation, breastfeeding, tobacco cessation in pregnancy, prevention and management of communicable and non-communicable disease and obesity in pregnancy, early childhood development, and

postpartum depression [13]. Since they work across all levels of care, from communities to hospitals, midwives are uniquely placed to provide essential services to women and newborns in the most difficult settings [14].

However, despite their worth, The State of the World Midwifery Report 2021 estimates that there is a current global shortage of 900,000 midwives globally [15]. This shortage includes both coverage [15] and a lack of midwives with the necessary skills to deliver services in all settings including in humanitarian settings [15, 16].

Health-care providers, face a variety of challenges in providing services in HFS contexts, including physical (such as a threat to life, violence, and risk of infection during pandemics), psychosocial (trauma, fear, and stigma), and professional (lack of an enabling environment, support systems, quality education and professional development opportunities) [17, 18]. Midwives thus face challenging living and working conditions and a lack of security for a mainly woman workforce, especially for community-based care requiring travel and displacements [15, 16]. Furthermore, midwives are often excluded from emergency, preparedness, and response planning, thus leaving vulnerable women and newborns without effective access to care [11].

The quality of care is also a key challenge in the provision of SRMNCAH services in HFS. Emerging data show that largely excess mortality that is amenable to health care is more attributable to poor quality of care than to lack of access to care [19, 20]. Improvements in coverage and quality at the same time are thus important to enhance the health and well-being of marginalised populations living in HFS [21, 22]. To this end, there is a need for a rights-based, respectful, culturally sensitive and people-centred approach to providing care to women and newborns, specifically the most vulnerable. The provision of accessible and quality midwifery services does not only need to be responsive to women's and girls' needs but also delivered in conditions that facilitate the satisfaction of midwives themselves.

However, there is a dearth of knowledge on the availability of midwifery services in HFS and the perceptions and experiences of women and midwives in receiving and providing SRMNCAH care in humanitarian contexts. There is a need to bridge this knowledge gap to inform the development of appropriate, gender-sensitive and people-centred policies, and humanitarian preparedness and response plans, placing quality midwifery services at the centre.

### Objectives

We aimed to conduct a mixed-methods analysis to:

1. Map and assess the coverage of midwifery services in humanitarian and fragile settings;

2. Document the experiences of women using essential SRMNCAH services in humanitarian and fragile settings;

3. Assess the facilitators and barriers in the provision of midwifery care in humanitarian and fragile settings.

## Methods

### Study design

A concurrent mixed methods approach was applied, using secondary analysis of quantitative and qualitative data sources [23]. The Partnership for Maternal, Newborn and Child Health (PMNCH) convened a steering group to guide this work that included key international

stakeholders from the International Confederation of Midwives (ICM), United Nations Population Fund (UNFPA), World Health Organisation (WHO), London School of Hygiene and Tropical Medicine (LSHTM) as well as primary researchers from the studies included in the qualitative analyses.

### Framework

This study was guided by the Midwifery Services Framework developed by ICM [24], providing an evidence-based approach to supporting efficient and quality midwifery services. This framework is specific to midwifery services and comprehensively addresses the content of care for women and newborns, building on the assessment of worker availability, accessibility, acceptability and quality (AAAQ), in addition to multidimensional conditions to strengthen midwifery services.

### Settings

Humanitarian settings and fragile states frequently overlap. To reflect this, the classification of HFS used in this paper is a composite index categorising countries across a spectrum of fragility with six groupings: highest fragility, very high fragility, high fragility, fragile, low fragility and non-fragile settings. This was based on the triangulation of data from five classifications of humanitarian and fragile states, that is, Organisation for Economic Co-operation and Development (OECD), the World Bank, the Fragile State Index, the INFORM Severity Index, and the United Nations Office for the Coordination of Humanitarian Affairs (UNOCHA) data on humanitarian response plans [25–29].

### Qualitative data

Using an existing systematic review of support systems and enabling environments for midwifery care in humanitarian and fragile settings, four primary qualitative studies on midwifery services with women and midwives in humanitarian and fragile countries were identified [18]. Studies that were included explored the experiences, perceptions, and utilisation of midwifery lead care by both women and midwives. Studies that were selected adopted a format of key informant interviews and focus group discussions and involved local and national researchers.

Authors from all four primary qualitative studies were contacted to request participation and inclusion in this study. The authors from three of the studies responded and the raw data transcripts from these studies were used to conduct the secondary analysis. These three studies were in Afghanistan [30], the Democratic Republic of Congo (DRC) [31], and South Sudan [32].

In addition, the steering group advised that the research team ensures there is a representation of various humanitarian contexts including refugee camp settings. Therefore, a literature review of data from refugee camp settings was undertaken which identified one further study from camp settings in Greece [33].

The characteristics of qualitative data sources are included in Table 1. Participants were all women or midwives able to provide experiences of midwifery services during pregnancy, delivery and postnatally in humanitarian settings. All participants provided written informed consent and were 16 years of age and older. All participants were selected through purposive sampling. A total of 71 individual interviews and 11 focus group discussions were used in this study comprising 150 participants in total. Data from 75 women were included; 30% of women were from Afghanistan (n = 23), and 69% of women were from refugee camps in Greece (n = 52). Data from 75 midwives were included; 72% of midwives were from the DRC (n = 54), 4% of midwives from Afghanistan (n = 3), 20% of midwives from refugee camps in Greece (n = 15), 4% of midwives from South Sudan (n = 3). Face-to-face qualitative interviews

**Table 1. Characteristics of qualitative data sources.**

| Study name | First Author. Year | Country | Method of data collection | Participants |
|---|---|---|---|---|
| 'Being a midwife is being prepared to help women in very difficult conditions': midwives' experiences of working in the rural and fragile settings of Ituri Province, Democratic Republic of Congo [31] | Baba, Theobald, et al. 2020 | Democratic Republic of Congo | Focus Group Discussion | 22 midwives in 3 focus groups |
| | | | Semi-structures interviews | 26 midwives 6 former midwives |
| "The midwife helped me, otherwise I could have died": women's experience of professional midwifery services in rural Afghanistan—a qualitative study in the provinces Kunar and Laghman [30] | Thommesen et al. 2020 | Afghanistan | Focus Group Discussions | 12 women in 2 focus groups |
| | | | Semi-structured interviews | 3 midwives 11 women |
| Midwives' experiences of cultural competency training and providing perinatal care for migrant women a mixed methods study: Operational Refugee and Migrant Maternal Approach (ORAMMA) project [33] | Fair et al. 2021 | Greece | Focus Group Discussions | 30 women in 5 focus groups 15 midwives in 1 focus group |
| | | | Semi-structures interviews | 22 women |
| Barriers Faced by the Health Workers to Deliver Maternal Care Services and Their Perceptions of the Factors Preventing Their Clients from Receiving the Services: A Qualitative Study in South Sudan [32] | Mugo et al. 2018 | South Sudan | Semi-structured interviews | 3 midwives |

were conducted and audiotaped with each participant or focus group and transcribed verbatim. Interviews were conducted in local languages and a translator was utilised as necessary.

## Data analysis

Data were analysed using framework analysis methods [34]. All data transcripts were read and re-read thoroughly to ensure familiarisation of the data. On the initial review and familiarisation of the raw data transcripts, an initial framework of initial themes and categories was developed with reference to the midwifery services framework (Table 2). This framework was representative of the data sets included and prevented the overlap of data.

The framework was applied to the transcripts for indexing. Emergent themes and primary categories were then added, which refined and developed the theoretical framework further.

**Table 2. Draft analysis framework.**

| Initial Theme | Initial Categories |
|---|---|
| Package of care | • Content of care provided |
| Organisation of SRMNCAH Services | • Availability of midwifery services<br>• Accessibility of midwifery services<br>• Acceptability of midwifery services<br>• Quality integrated midwifery care |
| Workforce | • Recruitment and retention<br>• Deployment<br>• Education and training<br>• Regulation of services |
| Enabling environment | • Non-staff resources<br>• Mentorship<br>• Peer support and community engagement<br>• Professional and career development<br>• Safety of the environment<br>• Coordination of care |
| Adaptation to setting | • Ongoing monitoring and evaluation<br>• Adaptation to local needs and situation |

Following the refinement of the theoretical framework, charting and descriptive synthesising of the data occurred and enabled the merging of themes and sub-themes.

### Quantitative data

The primary determinant utilised was the overall density of midwives for HFS countries. Data from the WHO National Health Workforce Accounts (NHWA) were utilised to provide the coverage of midwives within humanitarian and fragile settings [35]. The NHWA, coordinated by WHO, is a system by which countries progressively improve the availability, quality, and use of data on the health workforce through monitoring of a set of indicators to support the achievement of Universal Health Coverage (UHC), Sustainable Development Goals (SDGs), and other health objectives.

In addition, point estimates for the midwifery density, skilled birth attendance, and birth rate were provided for the three countries of focus (Afghanistan, Democratic Republic of Congo (DRC) and South Sudan) that were used for the qualitative analysis component. The midwifery density estimates and skilled birth attendance were taken from the NHWA. The Birth rate was not provided within the NHWA but was found within the World Bank country estimates [35, 36]. The years for the data sets included for the point estimates for these three countries of focus were those that were closest to the time of the qualitative data collection in each of the three countries.

### Data analysis

Data were analysed and presented descriptively on the density of midwives, disaggregated by regions and levels of fragility. This analysis was based on the most recent NHWA data set from each country across the period of 2010–2018.

### Convergence of qualitative and quantitative data

Data were analysed in parallel, equally, and independently. The qualitative and quantitative results were interpreted together.

### Ethics approval

Each of the studies included in this paper received ethics approval from relevant regulatory bodies both at the research institute and country levels. The ethical approval for the ORAMMA project in Greece was obtained from Elena Venizelou—Alexandra General Hospital (Reference Number: 5154/12-03-2018, G.H. "Elena Venizelou—Alexandra") and from Sheffield Hallam University (Converis ER5851022). The ethical approval for the qualitative study from Afghanistan was obtained from the Regional Ethics Committee for Medical and Health Research in Norway (REK Vest 2017/301, approved April 4, 2017), as well as by the Institutional Review Board at the Ministry of Public Health in Afghanistan. Ethics approval for the study from DRC was granted by the Liverpool School of Tropical Medicine (Research protocol 17–024) and the Multidisciplinary Research Centre for Development in Bunia, DRC (018/2017). Finally, for the study from South Sudan, the ethical review committee of the Department of Policy, Planning, Budgeting and Research of the Ministry of Health, Government of South Sudan, Juba, the Republic of South Sudan reviewed and approved the study.

## Results

### Quantitative results

An initial mapping of geographic distribution and level of fragility was conducted (Fig 1). The highest number of countries experiencing some level of fragility was noted in sub-Saharan Africa.

An analysis of the distribution of midwifery density across levels of fragility was conducted. The median density of midwives was noted to be highest in settings of no fragility (Fig 2).

A focussed mapping of midwifery density across humanitarian and fragile setting countries was conducted (Fig 3). It was noted that the poorest density was noted in sub-Saharan Africa.

Point estimates for the density of midwifery, skilled birth attendance and crude birth rate across all three priority countries were highlighted in Table 3. The point estimates highlight that all three countries were noted to be either the highest or have very high fragility levels. Crude birth rates were similar across the three settings and births are not universally attended to by skilled health personnel in all three settings. In 2018, the ratio of midwives was 1.39 per 10,000 population in Afghanistan and in 2016, 0.21 per 10,000 population in the DRC [35].

### Qualitative results

**Experiences of women in receiving SRMNCAH care.**   Data from Afghanistan and camp settings in Greece provided knowledge of the experiences of women receiving care. Data from all four settings highlighted the experiences of midwives in providing high-quality care.

### Package of care

Mothers consistently described midwives as being primary health-care practitioners providing comprehensive care across the continuum of pregnancy and birth including antenatal,

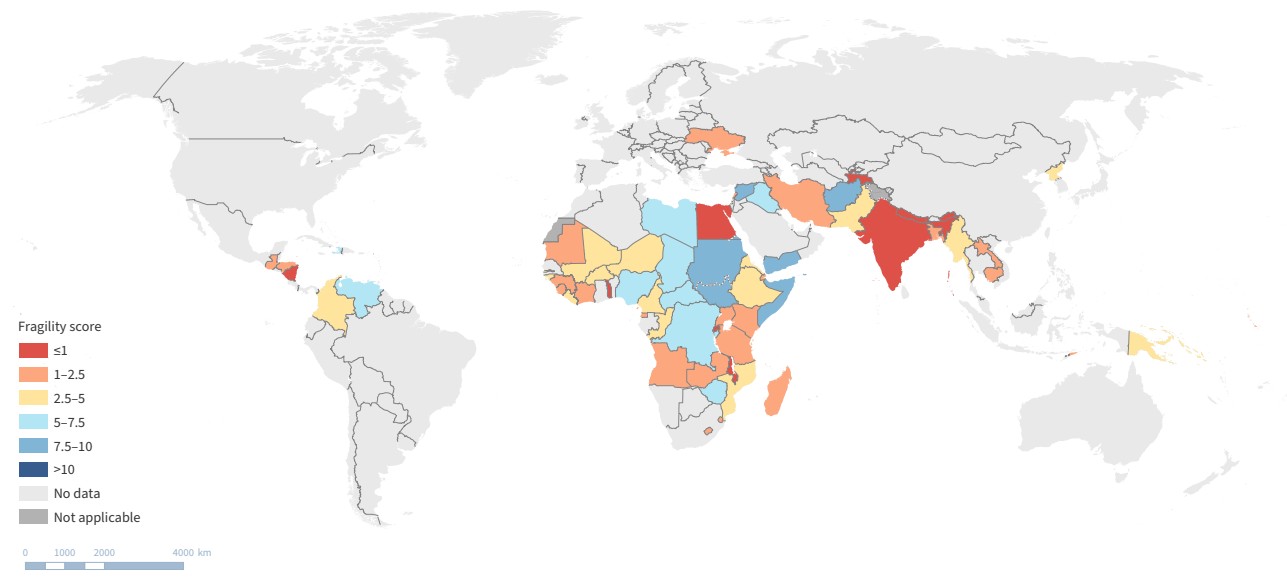

**Fig 1. Geographic distribution and level of fragility.**

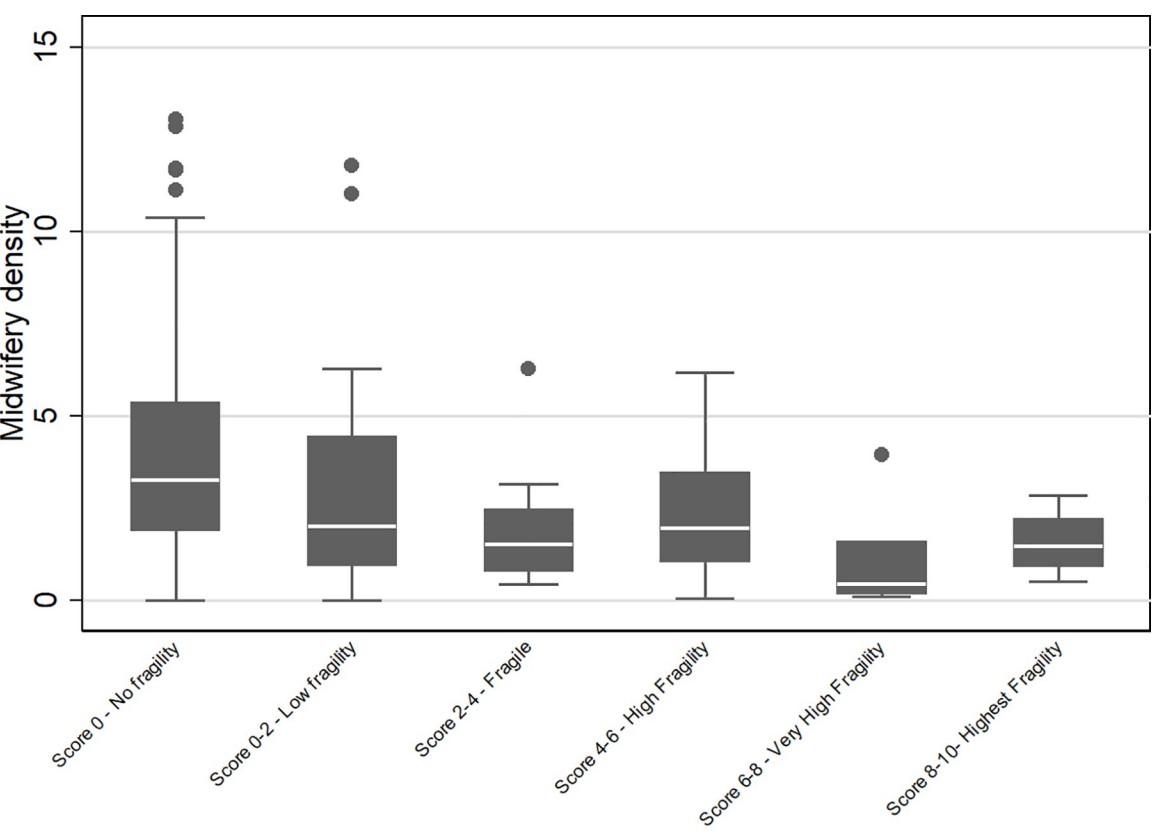

**Fig 2. Distribution of midwifery density across levels of fragility.**

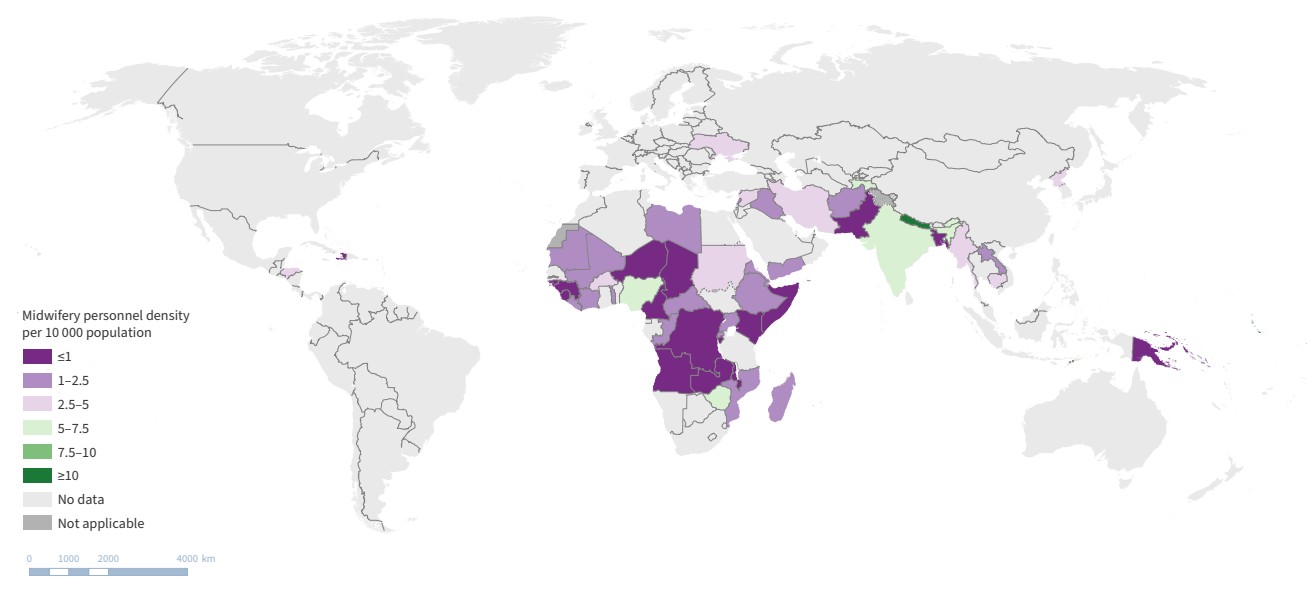

**Fig 3. Midwifery density in HFS countries.**

**Table 3. Point estimates from the three priority countries.**

|  | Afghanistan | DRC | South Sudan |
|---|---|---|---|
| Midwives per 10,000 [35] | 1.39 (2018) | 0.21 (2016) | N.R |
| Number of midwives [35] | 5,098 (2018) | 1793 (2016) | N.R |
| Births attended by skilled health personnel [35] | 59% (2018) | 80% (2014) | 39.7% (2020) |
| Crude Birth Rate per 1,000 [35] | 37 (2018) | 43 (2016) | 36 (2015) |
| OECD [25] | Extremely fragile | Extremely fragile | Extremely fragile |
| FSI 2020 [29] | High alert | High alert | Very high alert |
| INFORM Severity (Oct 2020) [27] | Very high | Very high | Very high |
| WB FY21 [26] | High-intensity conflict | Medium-intensity conflict | High-intensity conflict |
| Fragility level | Highest fragility | Very high fragility | Highest fragility |

intrapartum and postnatal services. The content of the services provided included an assessment of the mother and baby and provision of essential services, including initial management of postpartum haemorrhage, counselling on nutrition and breastfeeding, provision of specific food packages, prevention of endemic disease for the general health and well-being of the mother, e.g., anti-malarial medication.

## Accessibility of midwifery services

Once mothers had decided to seek care, they reported several factors associated with their ability to access and utilise care. Taking an integrated approach to service delivery for mothers and babies was seen as an enabler to ensure rapid treatment of both mother and baby that was convenient for mothers especially when travelling long distances for health care. Conversely, fragmented services were perceived as a barrier to accessing maternal-newborn care.

> "Today, I went to clinic, vaccinator didn't vaccine my baby and said to me 'I busy, go to Jalal for vaccine'. I come back home." [Woman- Afghanistan]

The impact of conflict and concerns for their safety was reported by mothers and their families as an additional barrier preventing access to care.

Additionally, for many, the distance to reach facility-based midwifery services is extensive and the transport to reach health centres is often inconsistent and costly and an expense that some families could not afford.

> "I didn't like being asked to go to the hospital regularly because of the distance. Sometimes there were delays at the hospital and I had to wait a long time." [Woman- refugee camp, Greece]

Socio-cultural considerations were reported to play a significant role in the mothers' choice to seek midwifery care. Some mothers reported the need to request permission from their husbands and/or elder relatives to seek care, receive certain treatments or use money to seek care for themselves and their babies. Furthermore, mothers often felt most comfortable discussing their health with women health-care providers.

> "I don't feel comfortable talking about my health issues with men. I prefer female interpreters." [Woman- refugee camp, Greece]

### Availability of midwifery services

As a result of the insufficient midwifery workforce numbers, midwives were concerned about the safety and well-being of the women they serve. Several midwives reported feeling unable to leave facilities at the end of their shifts and pressured to work unmanageable hours.

*"In this clinic, we are 2 midwives, one night I am in the house and the other night I am in the clinic. My colleague midwife had a car accident, her hand had broken, I was 17 nights and days in clinic, I didn't go to the house."* [Midwife- Afghanistan]

Additionally, to meet the health demands of the population, midwives reported that they were increasingly providing additional health services alongside their existing clinical duties which competed for their time and contributed to their increasing workload. The high and growing workload was highlighted as a potent determinant for perceived disrespectful care.

### Acceptability and quality of midwifery services

Trust and respectful care were reported factors influencing the uptake of midwifery services. Sometimes, distrust of facilities and staff including midwives due to prior negative experiences or perceived judgement on cultural practices acted as an impediment to using midwifery services. In other settings, mothers valued the importance of safe, confidential and quality respectful midwifery care where they were treated with dignity and were therefore satisfied with their care.

*"Yes, I am very satisfied. I felt respected, I didn't feel judged and I trusted my midwife a lot. I was sure that anything we discussed would stay between us. This is very important for me."* [Woman- refugee camp, Greece]

With the increasing displacement of people and disruption to health systems and health centres due to conflict, greater action has been taken to strengthen community engagement and trust. In the DRC, midwives reported collaborating with traditional birth attendants and integrating them into health care delivery to provide a critical link to communities to provide quality care. These traditional birth attendants have a unique skillset in being trusted by communities and can effectively share knowledge and care in a language and framing most appropriate to be culturally acceptable to communities.

*"The best way of supporting midwives is the promotion of integration of traditional birth attendants in the health facility, so that they work together with midwives, of course after basic training. That will help to deal with workload problems, but also building a better rapport with the population."* [Midwife–DRC]

In camp settings in Greece, the midwives and women reported the benefits of having a maternity peer supporter to assist in the provision of quality respectful care. The maternity peer supporter was a person from the woman's community. The peer supporter aided the midwives by providing information and support to the women and their families in languages and ways that were culturally sensitive. This was reported to be particularly effective in a camp setting where women were displaced and in need of support from someone they trusted most.

### Enabling environment for midwifery services

Midwives reported having to work in unsafe environments which included working in conflict settings putting them at risk of violence, and working in endemic areas with a high risk of

disease without the resources to protect them. Midwives also reported working at night or in remote settings, increasing the chance of working alone, which especially made women midwives feel increasingly unsafe.

*"Another big challenge is insecurity. With all the conflict we are going through, people are afraid to go to work there; they want to be where their security is assured." [Midwife- DRC]*

Working in a safe and supportive environment was reported to facilitate midwives to work effectively. Strategies to ensure a safe working environment included a buddy system with other midwives on night shifts and the employment of guards at facilities to protect the health workers.

*"We requested a bodyguard or security person to be present at the centre at night and another one to travel with the ambulance when we are transporting the sick person at night. Because the car can be stopped by the robbery and we are in danger. Simply because all of us here are women and we don't have the power to fight if anyone attacks us. We only pray to God, so we can be safe at night." [Midwife-South Sudan]*

Midwives also consistently reported insufficient funding and remuneration for their work. Some midwives in DRC reported not receiving any salary from government bodies, especially those working in rural settings.

*"A midwife is a big personality in rural Ituri, they respectively call her "MUNGANGA" or « MUZALISA », she has a high consideration in the community. But at the same time, she seems to be the most neglected by the government. She works in very bad conditions. The government does not pay them; they do not benefit from support from the government, especially those who work in the public sector." [Midwife-DRC]*

The provision of adequate and consistent pay enabled midwives to continue their work and was a potent deterrent to midwives leaving their profession. Several midwives reported supplementing their income from locally generated income at the health facility and non-governmental facilities run by NGOs/CSOs and faith-based organisations. These facilities were able to provide financial remuneration for services as well as the relevant resources and equipment necessary to provide midwifery services.

In all humanitarian settings, midwives reported feeling neglected and undervalued by governments. Concerns were shared about governments not recognising the valuable role midwives play in supporting women and their families and prioritising other healthcare cadres.

*"I also think that another challenge is about the consideration of midwives by the government. It seems like the government does not take our work seriously. It only minds about doctors, and sometimes nurses, but we are seen as those of the third category. For me, not being considered at the same level as doctors and nurses, I see it as a challenge." [Midwife-DRC]*

As a consequence of not being valued, midwives reported difficulties in attaining leadership roles in favour of alternate health-care worker contemporaries. This was an emerging theme identified.

Collaborative working across cadres was reported as an effective strategy to provide comprehensive quality care with each cadre understanding their roles and responsibilities.

*"As far as professional relationship is concerned, we are in a very good relationship with midwives, nurses, and the doctor we have here. If we face some difficulties that require their interventions, they are ready to come and help us. We are really in good relationship with all of them."* [Midwife-DRC]

## Recruitment and retention of the midwifery workforce

Poor working conditions were widely reported as impacting the potential of midwives to provide good care. Consistently midwives across settings reported a lack of enabling infrastructure e.g., lack of electricity, poor water supply, lack of referral transport, insufficient bed number and space as well as a lack of clinical resources including medication and relevant investigations necessary to deliver quality care. As a consequence of the difficulties in their work settings, midwives reported not responding to more complex conditions and medical complications.

Consistently midwives reported having an increasingly high workload in conflict-affected settings. This is an emerging theme identified. Midwives reported needing to respond to larger volumes of women especially those displaced due to conflict. As such, women may not speak the same language as the midwives providing care, thus adding difficulties in gaining a woman's trust and providing compassionate and quality care.

*"There are quite a few challenges when working with women in refugee camps, the biggest of which is gaining their trust. It is hard especially for a young aged midwife to earn the trust of an older pregnant woman who doesn't speak your language."* [Woman- refugee camp, Greece]

Midwives reported a failure of governments and health facilities to recruit and retain midwives sufficiently to meet the growing demand. Poor recruitment was especially reported in rural areas due to fears for staff to work in areas they may not know, under poor working conditions without sufficient resources and equipment and often without pay.

*"Working in rural and remote areas is like being sent to work in hell with the devil."* [Woman- DRC]

Through the increased displacement and fragmentation of care, there was an increasing need for midwifery services in rural settings. Beyond payment, midwives reported that specific incentives and social amenities were invaluable and conducive to working in such settings. These included having additional land for farming to enable the acquisition of a small side business and pay to make up for the lack/poor salary, provision of childcare amenities to enable midwives to have the flexibility to conduct shift work without a detrimental impact on family, and provision of housing on hospital site to prevent long-distance travel to and from the facility.

Furthermore, midwives surveyed in humanitarian settings reported wanting to quit the health workforce. Factors associated with midwives deciding to leave midwifery included, the detrimental impact of the service provision on their own physical and mental health and wellbeing. This is due to the long physically demanding shifts responding to large volumes of emotionally complex and stressful clinical cases, incompatibility of the working hours and shift patterns with responsibilities at home, and poor pay insufficient to meet living requirements.

Across settings, the key facilitator to providing quality care for women and their families was having a sufficient number of trained midwives.

In general, the midwives interviewed reported having great pride and personal value for their profession. Several midwives reported that they had wanted to become a midwife since childhood and highlighted the respected and honoured role midwives have in the community. Midwives were also in tune with the unique and critical role of having to care for mothers and their families. Fostering the personal value midwives have for their job was reported as an incentive for midwives to join the speciality.

*"In regards to why I became a midwife, you know my mother was a traditional birth attendant, and she really wanted me to be trained as a midwife. I was also seeing her, the way she was caring for pregnant women, the way she was respected and honoured in the community, I also said to myself that I would like one day to be a midwife, and that wish met with my mother's dream, that is why I went to study midwifery." [Midwife-DRC]*

### Adaptation of midwifery services to local needs

The needs of women and families in conflict settings are unique and midwives in camp settings reported the need for specially tailored guidelines to effectively provide quality care. Mentorship and supportive training were reported particularly to support newly qualified midwives in managing clinically complex cases. With poor staffing numbers and the displacement of midwives in rural settings, it was not always possible for junior midwives to receive adequate support. Regular in-service training and formal revalidations were noted as key mechanisms to ensure the validation of skills. Additionally, formal supervision provided the opportunity for guidance in career development.

*"To be experienced, you need to be well trained, and you also need to have enough time on the maternity ward during your training, and be coached by those whose passion has always been on the maternity ward. Of course, I agree that the more cases you deal with, the more experience you get, but there is a need of having covered the preliminary, which I have just mentioned earlier. To get experience, one needs to have a great mentor." [Midwife-DRC]"*

## Discussion

This mixed methods study aimed to provide an overview of the midwifery density in humanitarian and fragile settings and a synthesis of the experiences of women receiving midwifery care, and barriers and facilitators for midwives providing care in humanitarian and fragile settings.

Our quantitative results show that sub-Saharan Africa accounts for the highest levels of fragility, outlining the critical need for SRMNCAH services provided by midwives. The region also accounts for the lowest density of midwives, thus highlighting a critical gap in the provision of essential services for vulnerable women, children, and adolescents. The quantitative analysis in the three priority countries highlights that in the highest fragility settings, the crude birth rate is more than double that of global crude birth rates at the time period measured, yet skilled birth attendance in these three settings is not universal. Furthermore, midwifery ratios, when measured, were far below global estimates. For example, in Afghanistan and the DRC, midwifery ratios were noted to be 1.34 and 0.21 per 10,000 respectively, significantly lower than the UK 2018 estimates of midwifery and nursing ratio of 8.4 per 10,000. Interestingly, in the DRC where skilled birth attendance was high at 80%, midwifery ratios remained critically low at only 0.21 per 10,000. Our findings align with the existing knowledge of the global

shortage of midwives. The State of the Midwifery Report 2021, reported that an additional 900,000 midwives are required to provide global SRMNCAH services by 2030. Our study has highlighted that there is an urgent need for governments to recognise the invaluable role midwives play and as such prioritise midwives in health workforce planning, especially in humanitarian and conflict settings. It is essential to not only recruit but retain existing midwives. Strategies suggested in our study for the retention of midwives included government investment and prioritisation through targeted policy and programmatic measures for example through the provision of social amenities and incentives.

The study has highlighted that a lack of finances is a potent barrier to access and provision of essential SRMNCAH services. The lack of finances was cited as preventing mothers and communities from being able to access essential SRMNCAH services as well as for midwives to provide services effectively. The financial barriers were highlighted worse for those most vulnerable from poorer and/or more rural and remote communities where the need is often greatest. Focussed attention to address this barrier is critical. From a user perspective, implementation of innovative financing mechanisms is critical to effectively address the barrier complemented by the necessary appropriate stewardship [37]. Focus must be taken on reducing the level of out-of-pocket expenditure to ensure equity in access, which is necessary for the fullest realisation of Universal Health Coverage. From a provider perspective, earmarked adequate and continued remuneration for services despite the conflict must be maintained to ensure consistent and fair pay for existing midwives.

Where midwifery services are available, a common thread reported across all settings examined was the need to optimise midwifery working conditions in humanitarian and fragile settings. This would enable midwives to function at their full capacity and provide high-quality care. Across settings, midwives reported a lack of sufficient resources in consistent supply that was available to them to conduct their work. Additionally, in such settings, as a consequence of ensuing crises, the overall workload and complexity of the cases being seen by the midwives are increasing too. There is an inherent dichotomy where midwives are faced with an increase in the complexity of conditions and cases to manage yet insufficient resources to address these challenges effectively. Furthermore, coping strategies may result in an increased risk of missed opportunities for the provision of integrated care. As such there is a great need for improved in-service training adopting capacity-strengthening approaches that include continued mentorship, supportive training, and supervision. In addition, better regulatory and professional accountability mechanisms need to be established and adhered to consistently.

Fears about the safety to use and provide care were reported by both mothers and midwives. Whilst this is of concern in all settings, it is especially important in humanitarian and fragile settings, where the risks to safety and security are more frequent. This barrier impacts both the supply and demand sides of midwifery services. Specific measures identified in the studies included ensuring a safe and secure place for midwives to work, providing security guards at facilities, and operating a buddy system, particularly at night [18]. Actions must be taken at every health system level to ensure both users and providers are safe, which would in turn lead to greater acceptability of services.

The study supports the critical interconnections between the individual domains within the midwifery services framework. Midwifery services that are adapted to local needs in humanitarian and fragile settings and mediated by trust are intrinsically linked to the acceptability, availability and accessibility of services. Complex adaptive health system approaches lie at the heart of the Sustainable Development Goals and are critical to achieving targets by the 2030 deadline. In adopting a multi-sectoral lens to address these barriers, there is a concerted need for health systems investments to strengthen the midwifery cadre in humanitarian and fragile settings, including recruitment, deployment, and retention. Investments are required to

improve the midwifery ratio and availability of services, in addition to strengthening the enabling environment and support systems, which are critical to addressing the complex needs of midwives including psychosocial support.

The lack of midwifery in health workforce leadership positions was reported in the studies as a key barrier to strengthening the midwifery workforce. A 2016 report that documented the voices and experiences of 2,470 midwifery personnel across the world noted that while midwives are deeply committed to providing the best quality of care, they are also frustrated by not being involved in decision-making processes [38]. The report highlighted that it is not simply about fixing financial resources or health systems, but about redressing complex hierarchies of power and transforming gender dynamics. Respondents highlighted that "power, agency, and status" are important for midwives if progress is to be made in providing quality care [38]. As such, there is a great need for investment in midwifery leadership and governance by creating senior midwifery positions and strengthening institutional capacities and opportunities for midwives to drive health policy advancement [11].

On numerous aspects, the barriers to the uptake of essential SRMNCAH services provided by midwives are similar in resourced-strained settings, yet the scale of the challenges is greater in HFS. The experiences of mothers highlighted the essential value of community engagement, trust, and respectful care in ensuring the uptake of essential SRMNCH services and continuity of care in humanitarian and fragile settings. Examples of disrespectful care are documented in all settings but are especially dire in HFS, considering the difficult conditions in which midwives operate. There is therefore a need for better governance and accountability to ensure the delivery of quality SRMNCAH services including in humanitarian and fragile settings. Additionally, cross-cadre collaboration with midwives working side-by-side with traditional birth attendants was reported by midwives as a key strategy to ensure that the care provided is woman-centred, respectful, and culturally sensitive. Furthermore, a study from Greece reported the value of empowering community members. Involving maternity peer supporters (MPS) and mothers was highlighted as critical to ensuring that access issues are understood and addressed and that opportunities for self-care and health improvement are optimised in humanitarian and fragile settings. These mechanisms speak to the larger role of strengthening primary health-care platforms in HFS.

### Strengths and limitations

The utilisation of multiple data sets globally optimised the rigour of the overall data set included in the study. To mitigate the risk of misinterpretation during the re-analysis, the first author of each original data set was invited to review the re-analysis and provide feedback to ensure the findings were truly reflective of the data. This added to the rigour of the analysis.

### Limitations

Firstly, the association between midwifery ratios and SRMNCAH outcomes is only at the ecological level and needs to be interpreted accordingly otherwise it would be at risk of ecological fallacy. Secondly, there was a lack of existing qualitative data from a variety of different humanitarian and fragile settings. Thirdly, careful efforts were made to ensure that the quantitative point estimates provided for the midwifery density linked to the data collection period for the qualitative data. However, for South Sudan, although the qualitative data collection period was 2015–2016, the closest quantitative estimate was 2020. The authors do not feel this significantly impacted the data if anything, given the lower amount in 2010, the estimate included in the analysis would be higher than the reality of this point estimate for the density of midwifery at that time. Fourthly, authors were requested to provide their data transcripts as they related to

midwifery services. As the primary studies were on broader topics, the transcripts included were reliant on the transcripts which the authors had shared. Finally, for the studies included in the qualitative analysis, there was a lack of clarity on whether midwives were defined according to ICM standards in all the settings vis a vis what each country recognizes as a registered midwife.

### Future work

The following future work would be critical:

### Research

- Contextualised data and evidence on the experiences of midwives providing care and women receiving care specifically in humanitarian and fragile settings.

### Clinical practice

- Improved in-service training adopting capacity-strengthening approaches that include continued mentorship, supportive training, and supervision

- Promotion of midwifery leadership and governance by creating senior midwifery positions and strengthening institutional capacities and opportunities for midwives to drive health policy advancement

- Focus on empowerment and capacity strengthening community members.

### Policy action

- Urgent government prioritization of midwives through improving recrtuiment and ensuring focus on retention too

- Reduction in out-of-pocket expenditure to ensure equity in access, which is necessary for the fullest realisation of Universal Health Coverage.

- Earmarked adequate and continued remuneration for services despite the conflict must be maintained to ensure consistent and fair pay for existing midwives.

- Focus on creation of an enabling and safe environment for both provider and user

- Strengthening primary health-care platforms in HFS.

### Conclusion

Midwives are critical in protecting the health and well-being of mothers and newborns. However, the value of midwifery extends across the reproductive spectrum in being able to provide full sexual and reproductive health coverage, linking communities to essential SRMNAH services, while accelerating the human rights agenda. Midwives are more likely than other SRMNAH workers to be posted, remain and continue to provide essential services in humanitarian and conflict settings, often at great risk to their safety. There is therefore a great need to understand the critical gaps and way forward to prioritise and protect midwifery, especially in humanitarian and fragile settings

Our study highlights that there is a critically low density of midwives in humanitarian and fragile settings and the need for midwifery services to be protected and prioritised through improved leadership, continuous education and training, accountability and governance, and focussed attention where the need is greatest. It is also important that midwives, like other practitioners in humanitarian and fragile settings, understand the rights and entitlements of women and girls, and the communication between the patients and the midwives is improved. In this regard, the provision of accessible quality midwifery services that are responsive to the needs and wants of women and girls should be part of preparedness and response plans, as well as humanitarian strategies, and should inform policies related to the composition, development, funding and distribution of the health workforce in HFS.

## Acknowledgments

The Partnership for Maternal, Newborn and Child Health (PMNCH) has contributed to the development of this work, as part of its 2021–25 Strategy and related workplans. PMNCH is the world's largest alliance for women's, children's and adolescents' health and well-being, with over 1,400 partner organizations working together through 10 constituency groups. Its work is funded by a range of government and philanthropic donors. More information can be found on https://pmnch.who.int/".

The authors would like to express their sincere gratitude to members of the steering group that have contributed to the publication of this research paper. These members included Frances McConville (World Health Organization), Franka Cadee (International Confederation of Midwives (ICM), Helga Fogstad (Partnership for Maternal, Newborn and Child Health), Neha Singh (London School of Hygiene and Tropical Medicine), and Sarah Bar-Zeev (United Nations Population Fund).

## Author Contributions

**Conceptualization:** T. Dey, M. G. Shah, E. V. Langlois.

**Data curation:** T. Dey, A. Baba, N. Mugo, V. Vivilaki.

**Formal analysis:** T. Dey, M. G. Shah, M. Boniol.

**Investigation:** T. Dey, A. Baba, N. Mugo, T. Thommesen, V. Vivilaki.

**Methodology:** T. Dey, M. G. Shah.

**Project administration:** T. Dey, E. V. Langlois.

**Supervision:** T. Thommesen, N. Alam, D. Okoro, P. Tenhoope-bender, T. Triantafyllou, E. V. Langlois.

**Validation:** T. Dey.

**Visualization:** T. Dey.

**Writing – original draft:** T. Dey, M. G. Shah, A. Baba, N. Mugo, T. Thommesen, V. Vivilaki, M. Boniol, N. Alam, M. Dibley, D. Okoro, P. Tenhoope-bender, T. Triantafyllou, E. V. Langlois.

**Writing – review & editing:** T. Dey, M. G. Shah, A. Baba, N. Mugo, T. Thommesen, V. Vivilaki, M. Boniol, N. Alam, M. Dibley, D. Okoro, P. Tenhoope-bender, T. Triantafyllou, E. V. Langlois.

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
