## [Decision Letter · Decision Letter 0]

18 Dec 2023

PGPH-D-23-01467

Reproductive, maternal, newborn, child and adolescent health services in humanitarian and fragile settings: a mixed methods study of midwives’ and women’s experiences

Dear Dr. Dey,

Thank you for submitting your manuscript to PLOS Global Public Health. After careful consideration, we feel that it has merit but does not fully meet PLOS Global Public Health’s publication criteria as it currently stands. Therefore, we invite you to submit a revised version of the manuscript that addresses the points raised during the review process.

We look forward to receiving your revised manuscript.

Kind regards,

Kaveri Mayra

Academic Editor

Journal Requirements:

2. Please provide separate figure files in .tif or .eps format only and remove any figures embedded in your manuscript file. Please also ensure all files are under our size limit of 10MB.

3. Figs 1 & 3: please (a) provide a direct link to the base layer of the map (i.e., the country or region border shape) and ensure this is also included in the figure legend; and (b) provide a link to the terms of use / license information for the base layer image or shapefile. We cannot publish proprietary or copyrighted maps (e.g. Google Maps, Mapquest) and the terms of use for your map base layer must be compatible with our CC-BY 4.0 license. 

Additional Editor Comments (if provided):

Reviewers' comments:

Reviewer's Responses to Questions

**Comments to the Author**

1. Does this manuscript meet PLOS Global Public Health’s publication criteria? Is the manuscript technically sound, and do the data support the conclusions? The manuscript must describe methodologically and ethically rigorous research with conclusions that are appropriately drawn based on the data presented.

Reviewer #1: Yes

Reviewer #2: Yes

2. Has the statistical analysis been performed appropriately and rigorously?

Reviewer #1: I don't know

Reviewer #2: Yes

3. Have the authors made all data underlying the findings in their manuscript fully available (please refer to the Data Availability Statement at the start of the manuscript PDF file)?

Reviewer #1: No

Reviewer #2: Yes

4. Is the manuscript presented in an intelligible fashion and written in standard English?

Reviewer #1: Yes

Reviewer #2: Yes

5. Review Comments to the Author

Reviewer #1: PGPH-D-23-01467

Reproductive, maternal, newborn, child and adolescent health services in humanitarian and fragile settings: a mixed methods study of midwives’ and women’s experiences

Authors provide a timely and valuable review of experiences of midwives and service users in HFS settings.

I am honored to review this manuscript. It is well written and conceptualized. I write this review as a practicing midwife- working in a low resource, urban setting in the US and a researcher of midwifery care services and workforce issues that impact midwives caring for systematically and historically marginalized peoples.

Is there a empirical definition of HFS? Please provide. What constitutes HFS?

Pg 2: authors state “Considering the close link they have to the communities, midwives are also uniquely placed to provide a full range of SRMNCAH people-centred care services beyond immediate maternal and newborn care.12 This is especially important in HFS where health systems are broken or fragile and access is inconsistent. It is estimated that midwives can provide 90% of essential SRMNCAH services and as such are critical for optimum, cost-effective service delivery.13 In addition to preventing maternal and newborn deaths, quality midwifery care can also improve over 50 other health-related outcomes, including - sexual and reproductive health, immunisation, breastfeeding, tobacco cessation in pregnancy, prevention and management of communicable and non-communicable disease and obesity in pregnancy, early childhood development, and postpartum depression.14 Since they work across all levels of care, from communities to hospitals, midwives are uniquely placed to provide essential services to women and newborns in the most difficult settings.1”

While all of this promotes the values of midwifery care - my concern is that this expectation of this type of value driven care is not fully actionable in the face of acute structural and resource instability in HFS. The way it is stated may misconstrue this possibility. I am weary of communicating that midwifery can offer this level of service coverage in ALL contexts, midwifery can offer this value in its ideal form. Can authors make sure this is restated in this paragraph? I too often see midwifery care being promoted as a singular solution to poor outcomes when in fact we know that using the QMNC framework requires that the entire ecosystem be invested in. Suggest “WHILE…. Midwives can provide 90% of SRMH…. We caution that these may be severely restricted in HFS” – otherwise you reify the false notion that midwives can make something out of nothing.

Study Design

Authors could explicitly state which elements are quant and which elements are qual - this is not clear anywhere. “A concurrent mixed methods approach was applied, using secondary analysis of primary quantitative and qualitative data sources.” Do authors have any other studies or methods papers that use this type of methodology to cite? It is uncommon.

Please name the four countries first and then explain how data was obtained or used.

Convergence - can authors explain why Greece was not included in the quantitative synthesis?

Appreciative of the qualitative descriptive synthesis. Authors should state that this is a descriptive qualitative synthesis - not interpretive, as these are two distinct fields in the methodological space.

How are authors naming the participants in the quotations? What do these lettesr/numbers mean that are at the end of each quote? Describe to readers or simplify, type of participant and place - otherwise it is difficult to decipher.

It is also not clear who these midwives are? Are they midwives from the countries that they are serving, are they foreign trained aid workers?

Of note - please do not use female and women interchangeably - female denotes biological category, woman denotes social/gendered category. I suspect authors mean women when they write female. Please correct throughout.

The convergence of data is the weakest aspect of this analysis. In essence there is very little convergence as they are in separate categories and presented separately. In true convergence, authors make the connections between these two data types for readers clear and decipherable.

The discussion section is compelling. Would authors consider taking out the implications for practice and put those in a separate section? You have a future work section and It may be beneficial to take all the recommendations that are scattered throughout the discussion and make a bulleted list that would be in the implications/future directions section.

Reviewer #2: Congratulations to the team on this novel research! I consider this article to be worthy of publication. It is well written and well considered overall. The study is well crafted and appears to comply with the ethical concerns of compiling a complex and comprehensive international study. The rationale for the study and study objectives were clear. The results are well-presented and the implications of the study are teased out.

6. PLOS authors have the option to publish the peer review history of their article (what does this mean?). If published, this will include your full peer review and any attached files.

**Do you want your identity to be public for this peer review?** For information about this choice, including consent withdrawal, please see our Privacy Policy.

Reviewer #1: No

Reviewer #2: **Yes: **Maeve O Connell

---

## [Decision Letter · Decision Letter 1]

24 May 2024

PGPH-D-23-01467R1

Reproductive, maternal, newborn, child and adolescent health services in humanitarian and fragile settings: a mixed methods study of midwives’ and women’s experiences

Dear Dr. Dey,

Thank you for resubmitting your revised manuscript to PLOS Global Public Health. Your resubmission addressed the concerns raised by Reviewer 1.

After careful consideration, we feel that it has merit and is suitable for publication but needs some minor adjustments to fully meet PLOS Global Public Health’s publication criteria as it currently stands. 

In the resubmitted manuscript there is one concern. Please do address this.

**Acronym DRC appears in Page 4 in the 5^th^ Paragraph**: studies were in Afghanistan 34, the DRC 35, and South Sudan 36

Please give the full form of the acronym here 

**
Full form of DRC - Democratic Republic of Congo Appears in Page 5
**

Data from 75 midwives were included; 72% of midwives were from the Democratic Republic of Congo (n=54), 4% of midwives from Afghanistan (n=3), 20% of midwives from refugee camps in Greece (n=15), 4% of midwives from South Sudan (n=3).

**
Please note PLOS Global Public Health does not proof or copy edit the accepted manuscript.
**

Therefore, we invite you to submit a revised version of the manuscript that addresses the points raised during the review process.

We look forward to receiving your revised manuscript.

Kind regards,

Bijoya Roy, PhD

Academic Editor

Journal Requirements:

1. We have noticed that you have uploaded Supporting Information files, but you have not included a list of legends. Please add a full list of legends for your Supporting Information files after the references list.

Reviewers' comments:

Reviewer's Responses to Questions

**Comments to the Author**

1. If the authors have adequately addressed your comments raised in a previous round of review and you feel that this manuscript is now acceptable for publication, you may indicate that here to bypass the “Comments to the Author” section, enter your conflict of interest statement in the “Confidential to Editor” section, and submit your "Accept" recommendation.

Reviewer #1: All comments have been addressed

Reviewer #3: (No Response)

2. Does this manuscript meet PLOS Global Public Health’s publication criteria? Is the manuscript technically sound, and do the data support the conclusions? The manuscript must describe methodologically and ethically rigorous research with conclusions that are appropriately drawn based on the data presented.

Reviewer #1: Yes

Reviewer #3: Partly

3. Has the statistical analysis been performed appropriately and rigorously?

Reviewer #1: I don't know

Reviewer #3: Yes

4. Have the authors made all data underlying the findings in their manuscript fully available (please refer to the Data Availability Statement at the start of the manuscript PDF file)?

Reviewer #1: Yes

Reviewer #3: Yes

5. Is the manuscript presented in an intelligible fashion and written in standard English?

Reviewer #1: Yes

Reviewer #3: Yes

6. Review Comments to the Author

Reviewer #1: comments addressed.

Reviewer #3: Please find a document with review comments attached.

7. PLOS authors have the option to publish the peer review history of their article (what does this mean?). If published, this will include your full peer review and any attached files.

**Do you want your identity to be public for this peer review?** For information about this choice, including consent withdrawal, please see our Privacy Policy.

Reviewer #1: **Yes: **P. Mimi Niles

Reviewer #3: No

---

## [Editor Report · Decision Letter 2]

3 Jun 2024

Reproductive, maternal, newborn, child and adolescent health services in humanitarian and fragile settings: a mixed methods study of midwives’ and women’s experiences

PGPH-D-23-01467R2

Dear Dr Dey,

We are pleased to inform you that your manuscript 'Reproductive, maternal, newborn, child and adolescent health services in humanitarian and fragile settings: a mixed methods study of midwives’ and women’s experiences' has been provisionally accepted for publication in PLOS Global Public Health.

Best regards,

Bijoya Roy, PhD

Academic Editor
